# *BMPR2* Variants Underlie Nonsyndromic Oligodontia

**DOI:** 10.3390/ijms24021648

**Published:** 2023-01-13

**Authors:** Jinglei Zheng, Haochen Liu, Miao Yu, Bichen Lin, Kai Sun, Hangbo Liu, Hailan Feng, Yang Liu, Dong Han

**Affiliations:** 1Department of Prosthodontics, Peking University School and Hospital of Stomatology & National Center of Stomatology & National Clinical Research Center for Oral Diseases & National Engineering Research Center of Oral Biomaterials and Digital Medical Devices, Beijing 100081, China; 2Frist Clinical Division, Peking University School and Hospital of Stomatology & National Center of Stomatology & National Clinical Research Center for Oral Diseases & National Engineering Research Center of Oral Biomaterials and Digital Medical Devices, Beijing 100081, China

**Keywords:** BMPR2, nonsyndromic oligodontia, variants, genetics, BMP-SMAD1/5/8 signaling

## Abstract

Oligodontia manifests as a congenital reduction in the number of permanent teeth. Despite the major efforts that have been made, the genetic etiology of oligodontia remains largely unknown. Bone morphogenetic protein receptor type 2 (*BMPR2*) variants have been associated with pulmonary arterial hypertension (PAH). However, the genetic significance of *BMPR2* in oligodontia has not been previously reported. In the present study, we identified a novel heterozygous variant (c.814C > T; p.Arg272Cys) of *BMPR2* in a family with nonsyndromic oligodontia by performing whole-exome sequencing. In addition, we identified two additional heterozygous variants (c.1042G > A; p.Val348Ile and c.1429A > G; p.Lys477Glu) among a cohort of 130 unrelated individuals with nonsyndromic oligodontia by performing Sanger sequencing. Functional analysis demonstrated that the activities of phospho-SMAD1/5/8 were significantly inhibited in *BMPR2*-knockout 293T cells transfected with variant-expressing plasmids, and were significantly lower in *BMPR2* heterozygosity simulation groups than in the wild-type group, indicating that haploinsufficiency may represent the genetic mechanism. RNAscope in situ hybridization revealed that *BMPR2* transcripts were highly expressed in the dental papilla and adjacent inner enamel epithelium in mice tooth germs, suggesting that *BMPR2* may play important roles in tooth development. Our findings broaden the genetic spectrum of oligodontia and provide clinical and genetic evidence supporting the importance of *BMPR2* in nonsyndromic oligodontia.

## 1. Introduction

Nonsyndromic oligodontia is an inherited developmental anomaly affecting craniofacial organs and characterized by agenesis of six or more permanent teeth (excluding the third molars) with or without abnormal tooth morphology, affecting the maxillofacial function, orofacial aesthetics, and mental health [1]. The prevalence of nonsyndromic oligodontia is estimated to be 0.25% of Asian populations in China and 0.14% of Caucasian populations in North America, Australia, and Europe [2].

Genetics is widely recognized as a major factor responsible for developmental defects in the tooth germ. To date, only seven genes (*MSX1*, OMIM #106600; *PAX9*, OMIM #604625; *EDA*, OMIM #313500; *WNT10A*, OMIM #150400; *LRP6*, OMIM #616724; *WNT10B*, OMIM #617073; and *GREM2*, OMIM #617275) have been confirmed as causative factors associated with nonsyndromic oligodontia [3,4,5,6,7,8,9] and there is evidence to support the candidacy of seven genes (*AXIN2*, *EDAR*, *EDARADD*, *DKK1*, *BMP2*, *BMP4*, and *KDF1*) that may play a role in nonsyndromic oligodontia [10]. Despite considerable efforts, the genetic etiology of nonsyndromic oligodontia remains largely unknown, suggesting the existence of unidentified related genes and molecular mechanisms. Presumably, additional components in several defined key pathways, such as WNT, TGF-β/BMP, and Hedgehog, and FGF signaling, which are involved in tooth development, may be associated with nonsyndromic oligodontia pathogenesis [11].

The bone morphogenetic protein receptor type 2 (*BMPR2*; OMIM *600799) gene encodes a transforming growth factor-β (TGF-β) type II receptor for bone morphogenetic proteins (BMPs). BMPR2 binds to type I receptors and BMPs to form a tetrameric complex that initiates BMP/SMAD-mediated signaling and participates in embryonic development, vasculogenesis, and bone formation [12,13]. *BMPR2^−/−^* mice consistently exhibit severe pulmonary hypertension, skeletal disorders, and embryonic lethality [14]. In humans, variation in the *BMPR2* gene is the major cause of pulmonary arterial hypertension (PAH, OMIM #178600) and pulmonary venoocclusive disease 1 (OMIM #265450) [15,16]; however, no variation has been identified in individuals with oligodontia.

This study is the first to identify novel *BMPR2* variants in individuals with nonsyndromic oligodontia via whole-exome sequencing (WES). Moreover, we performed in vitro functional analysis and investigated the spatiotemporal expression of *BMPR2* in mouse tooth germ development stages. Our results indicate that *BMPR2* can be considered a novel pathogenic gene of nonsyndromic oligodontia.

## 2. Results

### 2.1. Clinical Findings and Variant Identification

Tooth phenotypes, family diagrams, and DNA sequencing chromatograms of the four pedigrees are presented in Figure 1. The predictions of pathogenic variants and ACMG classifications of the novel *BMPR2* variants are summarized in Table 1.

In a four-generation family (family #537) with inherited nonsyndromic oligodontia, the 18-year-old male proband (IV-2) had a congenital absence of six permanent teeth, two retained deciduous teeth, and one cone-shaped maxillary lateral incisor on the right side (Figure 1A–F). The mother of the proband (III-4) had a congenital absence of 12 permanent teeth (Figure 1G), while the father (III-5) had normal dentition. III-4 confirmed that her mother (II-7), grandmother (I-2), maternal aunts (II-3 and II-5), maternal uncle (II-2), and cousin (III-1) had nonsyndromic oligodontia (Figure 1H). Since no deleterious variants of known oligodontia-associated genes were found in WES, we focused on the other undetermined gene(s) in pathways related to tooth development. We identified a novel *BMPR2* heterozygous missense variant: c.814C > T; p.Arg272Cys (Table 1). By performing Sanger sequencing, the variant was validated in members with available DNA samples. The sequencing results revealed that the heterozygous variant segregated with the affected individuals, but not with the unaffected family members, suggesting an autosomal dominant mode of inheritance (Figure 1I). Moreover, we performed non-invasive examinations with echocardiography and electrocardiograms on the proband and his mother, and the results showed no abnormalities in the structures and systems of their hearts and pulmonary arteries (Appendix A). However, non-invasive examination has certain limitations; in the future, further invasive right heart catheterization will be necessary for the diagnosis of PAH. Bone mineral density measurement using dual-energy X-ray absorptiometry (DEXA) showed a decreasing trend in bone mineral density and an increasing risk of osteoporosis in both the proband and his mother (Appendix A). Notably, the impact of loss of BMPR2 in mice was previously reported as associated with bone formation [17], which is consistent with our findings in humans.

To further explore the prevalence of the *BMPR2* variants in unrelated individuals with oligodontia and evaluate whether *BMPR2* is a candidate pathogenic gene, we screened the whole coding region of *BMPR2* in a cohort of 130 individuals with undefined etiology. Two individuals were found to carry two additional novel *BMPR2* variants (Figure 1J–O), indicating that the detection rate of *BMPR2* in nonsyndromic oligodontia was 1.5%.

Proband #615, a 26-year-old woman, presented with twelve congenitally missing permanent teeth and three retained deciduous teeth (Figure 1J). A novel *BMPR2* de novo heterozygous missense variant, c.1042G > A; p.Val348Ile, was identified in this proband, whereas it was not detected in her parents with normal dentition (Figure 1K,L, Table 1). Proband #531, a 26-year-old woman, had agenesis of fifteen permanent teeth and one retained deciduous tooth (Figure 1M). She carried another novel *BMPR2* de novo heterozygous missense variant, c.1429A > G; p.Lys477Glu, while her parents showed normal dentition (Figure 1N,O, Table 1).

### 2.2. Bioinformatics and Conservation Analysis

The three novel variants were all located in the kinase domain of the BMPR2 protein, which was encoded by *BMPR2* exons 5–11 (Figure 2A,B). According to SIFT, PolyPhen-2, MutationTaster, and the standards of ACMG pathogenic classification, the *BMPR2* p.Arg272Cys variant was likely pathogenic and the *BMPR2* p.Val348Ile and p.Lys477Glu variants were pathogenic (Table 1). Multi-species alignments of the BMPR2 amino acid sequence showed that Arg272, Val348, and Lys477 were highly conserved during evolution (Figure 2C).

### 2.3. Three-Dimensional Conformational Analysis

Homology modeling of the structures of the kinase domain was performed to analyze the conformational effects of BMPR2 variants (Figure 2D–I). The missense variant Arg272Cys led to the basic residue Arg272 being substituted with a Cys, a neutral amino acid with a sulfhydryl group, which was considerably shorter than Arg, resulting in a significant conformational change in the β-sheet near the 272^nd^ residue (Figure 2D,D’,E,E’). The Val348Ile variant resulted in the residue Val348 being substituted with Ile, an amino acid with a longer side-chain than that of Val, which may affect the interaction of the 348th residue with the surrounding residues (Figure 2F,F’,G,G’). The Lys477Glu variant led to the basic residue Lys477 being substituted with a Glu, an acidic amino acid with a shorter side-chain than that of Lys, which may result in a conformational change in the α-helix near the 477th residue (Figure 2H,H’,I,I’).

### 2.4. Genetic Mechanism Explorations of BMPR2 Variants in Oligodontia

To evaluate the functional effects of the three *BMPR2* variants, we constructed the BMPR2-knockout 293T cells and established a model of a heterozygous *BMPR2* variant in vitro. Western blotting showed that in BMPR2-knockout 293T cells, endogenous BMPR2 expression at the protein level was reduced by approximately 90% (*p* < 0.0001; Figure 3A,B). Subsequently, successful expression of wild-type and all variant BMPR2-GFP fusion proteins was detected in transfected BMPR2-knockout 293T cells at the predicted weight of 141 kDa (Figure 3C). Functional analyses revealed that the expression of phospho-SMAD1/5/8 was significantly inhibited in BMPR2-knockout 293T cells transfected with Arg272Cys, Val348Ile, and Lys477Glu variant plasmids when compared to that in cells transfected with wild-type plasmids (*p* < 0.0001; Figure 3C,D).

To further explore the pathogenic mechanism of heterozygous *BMPR2* variants associated with oligodontia, we simulated the *BMPR2* heterozygosity found in oligodontia in vitro and confirmed the successful expression of heterozygotes via Western blotting (Figure 3E). Activities of phospho-SMAD1/5/8 in three heterozygous variant groups (transfected with 0.5 µg wild-type plasmid and 0.5 µg variant plasmids) were approximately 30% lower than those in the 1.0 µg wild-type plasmid-transfected group, while the activities were approximately 20% higher than those in the 0.5 µg wild-type plasmid-transfected group (*p* < 0.0001; Figure 3E,F), suggesting that haploinsufficiency may represent the genetic mechanism in *BMPR2*-associated nonsyndromic oligodontia.

### 2.5. Expression Pattern of Bmpr2 during Murine Molar Development

RNAscope in situ hybridization assay was used to reveal the precise spatiotemporal expression of *Bmpr2* in mouse first mandibular molars at serial developmental stages. *Bmpr2* was widely expressed in the dental epithelium and mesenchyme during the process of tooth development (E11.5–E16.5, Figure 4A–F). At the cap stage (E14.5), *Bmpr2* expression was detected in the enamel organ, dental papilla, and dental follicle (Figure 4D). Notably, at the bell stage (E15.5–E16.5), *Bmpr2* was highly expressed in the inner enamel epithelium, outer enamel epithelium, secondary enamel knots, and adjacent dental papilla cells, while it was less expressed in the stellate reticulum (Figure 4E,F).

## 3. Discussion

In this study, we identified three novel *BMPR2* missense variants, p.Arg272Cys, p.Val348Ile, and p.Lys477Glu, in an autosomal dominant family and two isolated patients with nonsyndromic oligodontia. Our study is the first to link the genetic importance of *BMPR2* to human oligodontia.

Although previous studies demonstrated the association of *BMPR2* variants with PAH, all the individuals with *BMPR2* variants in our study presented oligodontia with no PAH-associated symptoms. Using non-invasive examinations of echocardiography and electrocardiograms, no PAH-related abnormalities were found in proband #537 and his mother. Previous studies showed that heterozygous variations in the *BMPR2* gene are found in nearly 70% of families with heritable PAH [18] and in 25% of patients with sporadic disease [19]. PAH is more common in women (female:male ratio of 1.7:1) [20]. However, the penetrance of PAH is incomplete; only approximately 10–20% of individuals with *BMPR2* variants develop the disease during their lifetime, suggesting that PAH development is triggered by other genetic or environmental factors [21,22]. Therefore, *BMPR2* variants can be considered as risk factors in patients, suggesting that they may be susceptible to PAH in future.

The BMPR2 protein consists of extracellular, transmembrane, kinase, and C-terminal cytoplasmic domains [12]. BMPR2 kinase domain is crucial for BMP signal transduction and it participates in BMPR2 kinase phosphorylation, type I receptor (BMPR1) activation, activated receptor complex formation, and ultimately, SMAD1/5/8 phosphorylation. In this study, all the variants—Arg272Cys, Val348Ile, and Lys477Glu—caused different conformational changes in the BMPR2 kinase domain, demonstrating that the kinase domain represents a germline variant hotspot in nonsyndromic oligodontia and plays a crucial role in the pathogenesis of nonsyndromic oligodontia.

Moreover, according to an analysis of human disease genes, approximately 50% of the mutations in membrane receptors were dominant [23]. Genetic dominance commonly results from the dominant-negative effect or haploinsufficiency. The dominant-negative effect refers to a defective subunit adversely affecting the activity of the normal one, whereas haploinsufficiency describes a situation of reduced activity by a heterozygous null allele [24]. It is sometimes difficult to determine a priori whether haploinsufficiency or the dominant-negative effect is responsible for the phenotypes, particularly in the case of intragenic point mutations. In this study, we analyzed the expression of BMPR2 by Western blotting, and demonstrated that the inactivated BMP/phospho-SMAD1/5/8 signaling was caused by the novel *BMPR2* heterozygous variants, suggesting that the effects of *BMPR2* are allele dose-dependent. Further results showed that compared to the wild-type group, the activities of phospho-SMAD1/5/8 in heterozygous variant groups were reduced due to a heterozygous null allele, but were no less than in a normal allele group (simulated by the 0.5 µg wild-type group). These findings indicate that a haploinsufficiency effect contributes to the genetic mechanism of *BMPR2*-associated nonsyndromic oligodontia, rather than the dominant-negative effect.

When reviewing previous research, we found that *Bmpr2**^−/−^* mice died before E9.5, that is, prior to bone development [14], so skeletal or dental defects could not be assessed. Since *BMPR2* variants are broadly considered to be associated with PAH in humans, studies of mouse models frequently focused on the circulatory system. In recent years, many molecules in the BMP signaling pathway have been reported to be related to tooth development. Previous studies have reported that variants of *BMP4* are related to oligodontia in humans [25,26]. During mouse embryonic development, conservative Bmp signaling regulates odontogenesis via reciprocal epithelial–mesenchymal interaction. Disruption of Bmp signaling causes retardation of early tooth development [27]. Mice with *K14-Cre; Bmpr1a^fl/fl^* exhibit the retardation of tooth development, loss of posterior limbs, and hair defects [28]. Moreover, conditional ablation of *Bmp2* in enamel using *Osx-Cre; Bmp2^fl/fl^* transfection results in decreased enamel mineralization and thickness, which confirms the role of Bmp2 in the regulation of postnatal enamel formation [29]. Additionally, *Wnt1-Cre; Bmp4^fl/fl^* mice show an arrest in the development of mandibular molars at the bud stage [30]. However, the genetic role of *Bmpr2* in tooth development and oligodontia remains unknown. Using RNAscope analysis, we found that *Bmpr2* was highly expressed in the primary enamel knot, secondary enamel knot, inner and outer enamel epithelium, and adjacent mesenchymal cells. These data demonstrate that *Bmpr2* may regulate the epithelial–mesenchymal interactions and cell differentiation during tooth development. These results are basically consistent with previous findings on Bmpr2 protein expression in mice [31], and the precise expression of *Bmpr2* in our study complements earlier data at the mRNA level. However, the definite role of *BMPR2* in the regulation of tooth development needs to be further investigated in the future.

## 4. Materials and Methods

### 4.1. Recruitment of Subjects

A four-generation family with nonsyndromic oligodontia was recruited at the Department of Prosthodontics, Peking University School of Stomatology (Beijing, China) during medical treatment of the proband. No evidence of other general abnormalities was found in this family. Additionally, a cohort of 130 unrelated individuals with nonsyndromic oligodontia from our database was enrolled for the expanded genetic screening. All individuals confirmed the absence of tooth extraction or loss. Informed consent was obtained from all the participants. This study and associated research protocols were approved by the Ethics Committee of the Peking University School and Hospital of Stomatology (PKUSSIRB-202162021).

### 4.2. WES

Genomic DNA was isolated from peripheral blood lymphocytes using a DNA Whole-blood Mini Kit (BioTek, Beijing, China) following the manufacturer’s instructions. WES was performed by Angen Gene Medicine Technology (Beijing, China) to identify potential pathogenic gene variations. The selection criteria for pathogenic genes were as follows: (1) genes involved in orodental development were screened; (2) short insertions or deletions and sequence nucleotide variants, including nonsense, missense, and splicing variations, with a minor allele frequency ≤ 0.01 in population frequency databases, such as the Genome Aggregation Database (gnomAD, https://gnomad.broadinstitute.org (accessed on 13 March 2021)) [32] and single nucleotide polymorphisms (dbSNP, https://www.ncbi.nlm.nih.gov/SNP/ (accessed on 13 March 2021)); (3) the potential pathogenicity was predicted via bioinformatics analysis using MutationTaster (http://www.mutationtaster.org (accessed on 13 March 2021)), Sorting Intolerant from Tolerant (SIFT, http://provean.jcvi.org (accessed on 13 March 2021)), and Polymorphism Phenotyping v2 (PolyPhen-2, http://genetics.bwh.harvard.edu/pph2/ (accessed on 13 March 2021)), and then the screened variants were classified according to the 2015 American College of Medical Genetics and Genomics guidelines [33].

### 4.3. Sanger Sequencing and Genetic Screening

From the WES results, we identified a novel variant of *BMPR2* (RefSeq NM_001204), which was validated via Sanger sequencing and familial co-segregation. Subsequently, to further confirm the genetic role of *BMPR2* in oligodontia, gene screening was performed via Sanger sequencing using samples of 130 unrelated individuals with nonsyndromic oligodontia with undefined etiology in our clinical database. Polymerase chain reaction (PCR) was performed to amplify the coding region of *BMPR2* (primers and amplification conditions can be provided) using Taq PCR Master Mix (BioTek, Beijing, China). The PCR products were sequenced by Tsingke Biological Technology (Beijing, China).

### 4.4. Bioinformatic and Conservative Analysis

Predictions of the putative pathogenic *BMPR2* variants were performed using bioinformatic tools, including SIFT, PolyPhen-2, and MutationTaster. Conservative analysis was performed using the ClustalX 2.1 program based on the multiple alignments of BMPR2 amino acid sequences (NP_001195) among different species, which were obtained from the National Center for Biotechnology Information (National Institutes of Health, Bethesda, MD, USA).

### 4.5. Three-Dimensional Structural Modeling

Since the identified BMPR2 variants were all distributed in the kinase domain of wild-type BMPR2 protein, an optimal template was selected for the structural homology modeling of the kinase domain (from amino acid 203 to 504) using SWISS-MODEL (https://swiss-model.expasy.org (accessed on 21 February 2021)). Changes in the three-dimensional structure of BMPR2 protein were observed using PyMOL software (Molecular Graphics System; DeLano Scientific, Palo Alto, CA, USA).

### 4.6. Plasmid Construction

To construct the wild-type *BMPR2* plasmid, the full-length coding region (3118 bp) of the human *BMPR2* gene was introduced to the green fluorescent protein (GFP)-expressing pEGFP-C1 vector between the 5′-EcoRI and 3′-XhoI sites. The wild-type plasmid pEGFP-C1-BMPR2, as well as the three variant plasmids, pEGFP-C1-Arg272Cys, pEGFP-C1-Val348Ile, and pEGFP-C1-Lys477Glu, were synthesized at the Beijing Genomic Institute (BGI, Beijing, China), and the entire sequence of the constructs was confirmed by BGI.

### 4.7. Cell Preparation and Western Blot Analysis

To eliminate endogenous interference, we used the clustered, regularly interspaced short palindromic repeats (CRISPR)/CRISPR-associated protein 9 (Cas9) system to delete endogenous *BMPR2* from 293T cells obtained from Tsingke Biological Technology. Subsequently, the BMPR2-knockout 293T cells (BKOTCs) were transfected with 1 μg empty vector, 1 μg wild-type plasmid, 1 μg variant plasmids, 0.5 μg wild-type plasmid, or 0.5 μg wild-type plasmid, together with 0.5 μg of each variant plasmid, using Lipofectamine 3000 (Thermo Fisher Scientific, Waltham, MA, USA).

After 48 h of transfection, recombinant GFP was observed under a fluorescence microscope (Olympus BX51; Olympus, Tokyo, Japan), and the transfected cells were lysed using protein lysis buffer (Beyotime Technology, Shanghai, China) containing phosphatase inhibitor (Solarbio, Beijing, China). Western blot analysis was performed using cell lysates, and proteins were incubated with anti-GFP (Abcam, Cambridge, United Kingdom), anti-p-SMAD1/5/8 (Cell Signaling Technology, Danvers, MA, USA), anti-SMAD1 (Cell Signaling Technology), and anti-glyceraldehyde 3-phosphate dehydrogenase antibodies (Abcam). The proteins were visualized using a chemiluminescent ECL kit (Sigma–Aldrich Corp., St. Louis, MO, USA) and semi-quantified using Image J software (National Institutes of Health). Experiments were performed in triplicate. Data were analyzed with Student’s t test using SPSS (IBM, Armonk, NY, USA). Results were presented as the mean ± standard deviation (n = 3), and *p* < 0.01 was considered significant.

### 4.8. Tooth Germ Preparation and RNAscope In Situ Hybridization Analysis

To detect the temporal and spatial expression pattern of *Bmpr2* during early tooth development, timed-pregnant ICR mice (Department of Laboratory Animal Science, Peking University Health Science Center, Beijing, China) at the stages of embryonic (E) days 11.5 (E11.5)–E16.5 were euthanized. The embryonic heads were dissected, fixed in 4% paraformaldehyde for 24 h, dehydrated, embedded in paraffin, and serially sectioned (at a thickness of 5 μm) in the coronal plane. RNAscope Probe-Mm-*B*MPR2 (Advanced Cell Diagnostics, Newark, CA, USA) was designed to target the mouse *Bmpr2* mRNA using an RNAscope 2.5 HD detection reagents-RED kit (Advanced Cell Diagnostics, 322360).

## 5. Conclusions

In conclusion, we provide the first genetic evidence of the etiological correlation between *BMPR2* heterozygous variants and nonsyndromic oligodontia. The spatiotemporal expression pattern of *Bmpr2* during tooth development suggests that *BMPR2* may play a crucial role in tooth morphogenesis. Our findings further extend the panel of candidate genes available for the genetic screening of nonsyndromic oligodontia and provide new insights into the molecular mechanisms of nonsyndromic oligodontia.

## Figures and Tables

**Figure 1 ijms-24-01648-f001:**
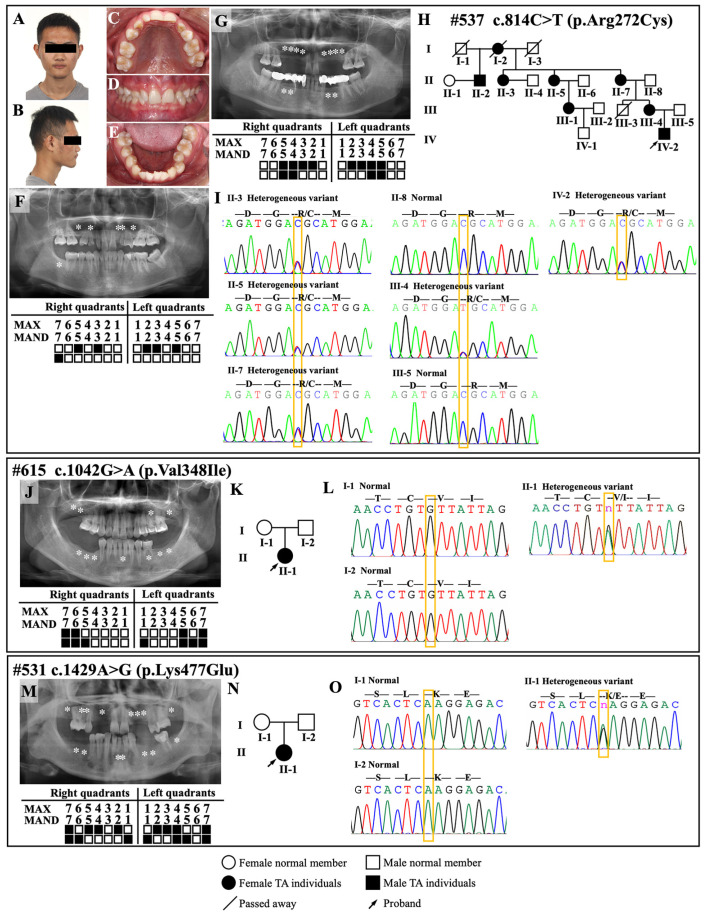
Clinical features and genetic screening of families with nonsyndromic oligodontia. (**A**–**G**) Digital photographs, panoramic radiographs, and missing tooth positions in proband #537 IV-2 (**A**–**F**) and his mother III-4 (**G**); (**H**,**I**) genogram and sequencing chromatograms revealed a novel heterozygous *BMPR2* variant (c.814C > T; p.Arg272Cys) identified in members with nonsyndromic oligodontia, IV-2, III-4, II-7, II-5, and II-3, which was inherited in an autosomal dominant pattern. Subjects I-1, I-2, I-3, II-1, II-2 II-4, II-6, III-1, III-2, III-3, and IV-1 were not available for clinical evaluation or DNA analysis; (**J**) panoramic radiograph and schematic of proband #615 II-1; (**K**,**L**) genogram and sequencing chromatograms show a de novo heterozygous *BMPR2* variant (c.1042G > A; p.Val348Ile) identified in the proband; (**M**) panoramic radiograph and schematic of proband #531 II-1; (**N**,**O**) genogram and sequencing chromatograms show a de novo heterozygous *BMPR2* variant (c.1429A > G; p.Lys477Glu) identified in the proband. The white asterisks (*) in the panoramic radiographs and solid squares in the schematics represent the missing permanent teeth. Mand, mandibular. Max, maxillary. The yellow arrows indicate retained deciduous teeth; the red arrows indicate cone-shaped permanent teeth. The Roman numbers on the left side of each genogram represent the numbers of generations in the family. *BMPR2*, bone morphogenetic protein receptor type 2.

**Figure 2 ijms-24-01648-f002:**
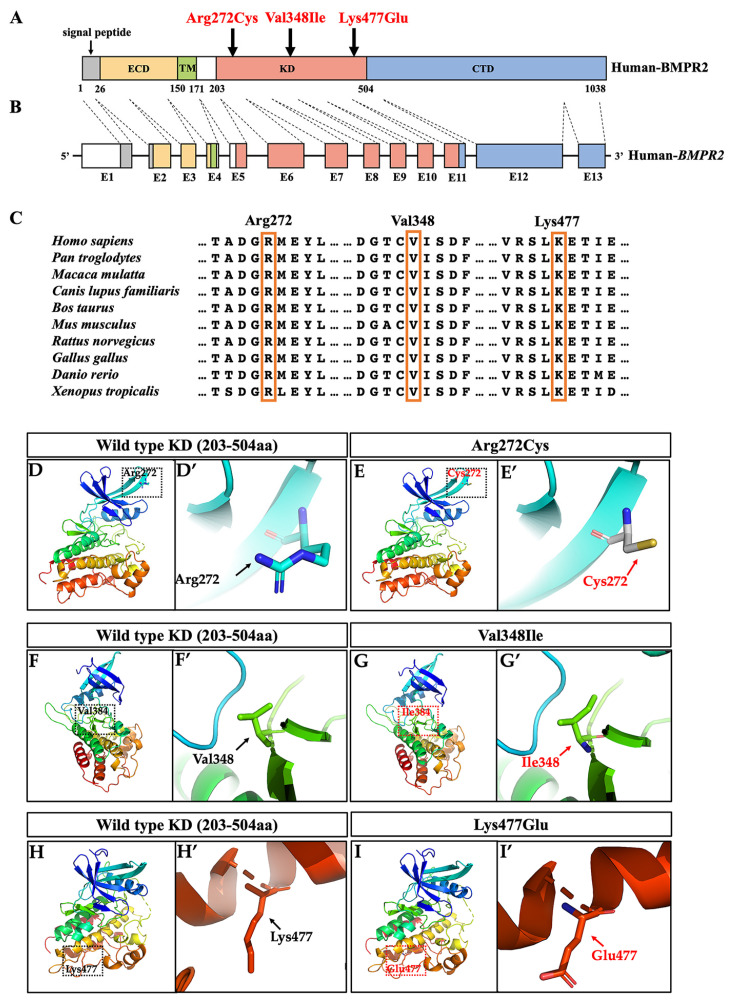
Location, conservation analysis, and tertiary structure analysis of BMPR2 variants. (**A**,**B**) Locations of three identified variants (marked in red) in schematic diagram of BMPR2. Correlations between exons and protein domains are shown in different colors. ECD, extracellular domain; TM, transmembrane domain; KD, kinase domain; CTD, cytoplasmic tail domain; (**C**) conservation analysis of the three variation sites among different species; (**D**–**I**) structural changes in three BMPR2 variants (**E**,**G**,**I**) compared with the wild-type BMPR2 (**D**,**F**,**H**) in the kinase domain. Dashed boxes denote the locations of Arg272 (**D**), Cys272 (**E**), Val348 (**F**), Ile348 (**G**), Lys477 (**H**), and Glu477 (**I**) residues; (**D’**–**I’**) higher magnifications of the boxed areas. aa, amino acid. BMPR2, bone morphogenetic protein receptor type 2.

**Figure 3 ijms-24-01648-f003:**
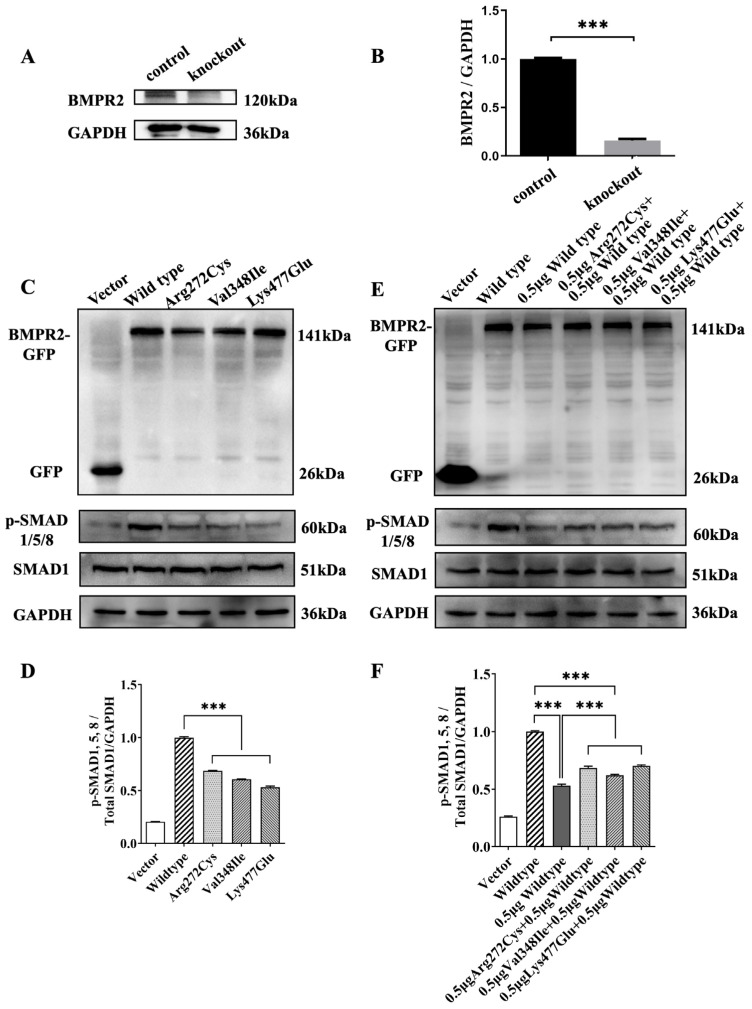
Genetic mechanisms of *BMPR2* variants causing oligodontia. (**A**,**B**) Western blot analysis of endogenous *BMPR2* gene knockout in 293T cells; (**C**) expression of BMPR2, p-SMAD1/5/8, and SMAD1 in wild-type and variant groups (Arg272Cys, Val348Ile, and Lys477Glu). An empty vector was transfected into cells as a negative control; (**D**) quantification of p-SMAD1/5/8 level via densitometry analysis using image J software; (**E**) expression of BMPR2, p-SMAD1/5/8, and SMAD1 in wild-type, 0.5μg wild-type, and three heterozygous variant groups; (**F**) quantification of p-SMAD1/5/8 levels by densitometry analysis using image J software. The asterisks denote the difference with statistical significance (*** *p* < 0.0001) compared to the wild-type group. The results are depicted as the mean ± standard deviation. BMPR2, bone morphogenetic protein receptor type 2.

**Figure 4 ijms-24-01648-f004:**
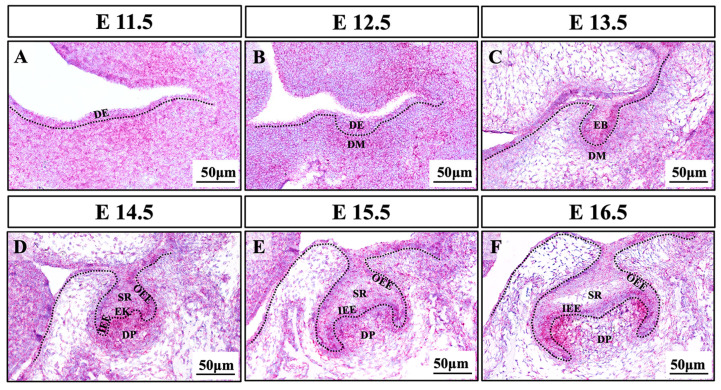
Expression patterns of *Bmpr2* during early tooth development in mouse. (**A**,**B**) Expression patterns of *Bmpr2* detected using RNAscope assay during the dental epithelium thickening stage (E11.5 and E12.5), (**C**) the bud stage (E13.5), (**D**) the cap stage (E14.5), and (**E**,**F**) the bell stage (E16.5 and E15.5). DE, dental epithelium; DM, dental mesenchyme; DP, dental papilla; DF, dental follicle; EB, epithelial bud; EK, enamel knot; IEE, inner enamel epithelium; OEE, outer enamel epithelium; SR, stellate reticulum; *Bmpr2*, bone morphogenetic protein receptor type 2. Scale bars: 50 μm.

**Table 1 ijms-24-01648-t001:** Harmful prediction of the variants in *BMPR2*.

Patient	Exon	Nucleotide Change	Protein Change	Variation Type	dbSNP	gnomAD (MAF)	SIFT	PolyPhen-2	Mutation Taster	ACMG Classification (Evidence of Pathogenicity)
#537	6	c.814C > T	p.Arg272Cys	Missense	rs773655445	0.00003186	0.008 (damaging)	0.999 (probably damaging)	Disease causing	Likely pathogenic (PS3 + PM2 + PP1 + PP3)
#615	8	c.1042G > A	p.Val348Ile	Missense	rs201067849	0.0005339	0.023 (damaging)	0.715 (possibly damaging)	Disease causing	Pathogenic (PS1 + PS2 + PS3 + PP3)
#531	11	c.1429A > G	p.Lys477Glu	Missense	— ^1^	— ^1^	0.060 (tolerated)	1.000 (probably damaging)	Disease causing	Pathogenic (PS2 + PS3 + PM2 + PP3)

^1^ Variant was not found in dbSNP or gnomAD; ACMG: American College of Medical Genetics; PS: pathogenic criterion is weighted as strong; PM: pathogenic criterion is weighted as moderate; PP: pathogenic criterion is weighted as supporting.

## Data Availability

The variants identified in this study were submitted to the ClinVar database (submission ID SCV002578204 and SCV002578205). WES data are available from the SRA database (accession number PRJNA900358).

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
