# Peer review of "BMPR2 Variants Underlie Nonsyndromic Oligodontia"

_ijms, 2023, doi:10.3390/ijms24021648_

Round 1
Reviewer 1 Report
Zheng et al. discovered a family (#537) with oligodontia (tooth agenesis). The inheritance pattern of this family was consistent with an autosomal dominant mode or haploinsufficiency. They identified a novel variant in BMPR2 by whole-exome sequencing. With this information, they sequenced a cohort of 130 individuals with undefined etiology and found two additional variants in BMPR2. The analysis of recombinant variants in cultured cells seems to suggest that their activities are reduced, compared to the wild-type. Based on this observation, they concluded that haploinsufficiency accounts for the inheritance.
This is a possibly interesting study. However, their analysis on cultured cells was not conclusive to this reviewer. Importantly, previous studies on this gene in mouse models and humans are not consistent with this study, raising a concern.
Figure 1I: Because the resolution of the sequencing chromatograms was poor in the review version I was not able to tell the heterozygocity.
Figure 3: BMPR2 KO cells transfected with the empty vector showed high levels of p-SMAD. The expression of wild-type BMPR2 did not enhance the amounts of p-SMAD compared to the empty vector control. As these controls are not behaving as expected, this reviewer is not able to interpret C-F. The authors used a transient transfection approach. It would be better to generate cells stably expressing the constructs and evaluate the downstream signaling with the cells.
What concerns this reviewer is that previous studies on this gene are not consistent with this study. Homozygous BMPR2 KO (BMPR-/-) died before E9.5 prior to bone development (Beppu et al., 2000 Dev Biol 221:249-258). So, skeletal or dental defects could not be assessed. Heterozygous BMPR2 KO (BMPR+/-) mice had a reduced birth rate but were morphologically normal and had the same life span as wild-type mice (Long et al. 2006. Cir Res. 98:818-827; Song et al. 2005. Cirulation 112:553-562). Obviously, the heterozygotes did not show any dental phenotypes. BMPR2 knockdown mice did not show any dental phenotypes (Liu et al. 2007. Blood 110:1502-1510). Clearly, gene dosage did not affect dental development in mice.
Hetrozygous mutatations in BMPR2 caused primary pulmonary hypertension in humans and dental elements were not implicated in this disease (2000 Nat Genet. 26:81-84; Rigelsky et al. 2008 AJMG 146A:2551-2556). Nature of these mutations include frameshift, nonsense and missense mutations.
Therefore, it is really difficult to reconcile the conclusion of this manuscript with previous studies.
Author Response
Dear reviewer1:
On behalf of our co-authors, we thank you very much for giving us the opportunity to revise our manuscript. We appreciate you for positive comments and constructive suggestions on our manuscript ijms-2039845 entitled “BMPR2 variants underlie nonsyndromic oligodontia”. We have studied all your comments carefully, and the modifications in the revised manuscript were in track mode. We hope this revision could merit publication considering the careful modifications that we made. The point-by-point responses are provided below.
Point 1: Figure 1I: Because the resolution of the sequencing chromatograms was poor in the review version I was not able to tell the heterozygocity.
Response 1: Thank you for your constructive suggestion. We totally agree with it and have improved the resolution in our revised Figures 1.
Point 2: Figure 3: BMPR2 KO cells transfected with the empty vector showed high levels of p-SMAD. The expression of wild-type BMPR2 did not enhance the amounts of p-SMAD compared to the empty vector control. As these controls are not behaving as expected, this reviewer is not able to interpret C-F. The authors used a transient transfection approach. It would be better to generate cells stably expressing the constructs and evaluate the downstream signaling with the cells.
Response 2: Thank you very much for your professional comments. Our molecular experiments had been repeated and the results was shown no significant difference in the expression of phospho-SMAD between the vector group and the wild-type group. The reason might be as follow. The molecular weight of BMPR2-GFP fusion protein was much larger than that of GFP tag, so the transfection efficiency was difficult to be consistent during transient transfection, resulting in strong background expression of the vector. For the construction of stable expression cell lines, we consider that transient transfection of mutant plasmids is a common experimental method for suggesting the mechanism of gene point mutation. And the effect of transient transfection was available when the protein was collected after 48 hours. We will follow your suggestion to construct stable-transfected cell lines in further experiments in case longer observation time is needed.
Point 3: What concerns this reviewer is that previous studies on this gene are not consistent with this study. Homozygous BMPR2 KO (BMPR-/-) died before E9.5 prior to bone development (Beppu et al., 2000 Dev Biol 221:249-258). So, skeletal or dental defects could not be assessed. Heterozygous BMPR2 KO (BMPR+/-) mice had a reduced birth rate but were morphologically normal and had the same life span as wild-type mice (Long et al. 2006. Cir Res. 98:818-827; Song et al. 2005. Cirulation 112:553-562). Obviously, the heterozygotes did not show any dental phenotypes. BMPR2 knockdown mice did not show any dental phenotypes (Liu et al. 2007. Blood 110:1502-1510). Clearly, gene dosage did not affect dental development in mice.
Heterozygous mutations in BMPR2 caused primary pulmonary hypertension in humans and dental elements were not implicated in this disease (2000 Nat Genet. 26:81-84; Rigelsky et al. 2008 AJMG 146A:2551-2556). Nature of these mutations include frameshift, nonsense and missense mutations.
Therefore, it is really difficult to reconcile the conclusion of this manuscript with previous studies.
Response 3: Thank you so much for your observation and professional comments. We have carefully read your suggestions and related literatures. We also found previous analyses about the embryonic lethal phenotype of Bmpr2 knockout mice, however, there were no reports of tooth phenotype in Bmpr2 knockout mice. Since BMPR2 mutations were considered to be associated with pulmonary hypertension in human, studies of mouse models frequently focused on the circulatory system. The tooth phenotype in the mouse model was likely to be ignored if only small changes occured. Moreover, the penetrance of PAH is incomplete: only approximately 10–20% of individuals with BMPR2 variants develop the disease during their lifetime, suggesting that different types and loci of BMPR2 variants may result in different phenotypes. In recent years, many molecules in the BMP signaling pathway, such as BMP receptor type 1 (Andl et,al., 2004, Development, 131:2257-68), BMP ligands BMP2 (Feng et al., 2011, Cell Tissues Organs, 194:216-21) and BMP4 (Yu et al., 2019, AOB, 103:40-46; Huang et al., 2013, EJOS, 121:313-8), have been reported to be related to tooth development. BMPR2 is a member of the BMP pathway, as well as widely expressed in mouse tooth germs, suggesting that it can be considered as a candidate molecule affecting tooth development. These contents have been reflected in the last paragraph of our “Discussion” and we also have expanded our discussion in the revised manuscript. In order to further explore the tooth phenotype, we have already constructed Bmpr2 conditional knockout mice. We also look forward to your continued attention to the experimental progress of our research.
Reviewer 2 Report
In the article entitled “BMPR2 variants underlie nonsyndromic oligodontia”, the authors broaden the genetic spectrum of oligodontia and provide clinical and genetic evidence supporting the role of BMPR2 in the etiology of non-syndromic oligodontia. The article is comprehensive and suitable for publication after some minor points are addressed:
Minor points:
1) In line 94-95, the authors state that their findings suggest that BMPR2 is involved in bone formation. However, this has already been shown previously (e.g., Lowery et al., 2015, J Cell Sci.). This sentence needs to be rewritten to incorporate these previous findings.
2) Text in Figures 1 and 3 are too small when figures are printed. The font size needs to be increased.
3) In lines 155-156, the authors use an acronym (BKOTCs) to describe the BMPR2 knockout cell line. This was confusing at first because this acronym was not described previously. I would rather write down everything without using an acronym.
4) In lines 172-173, the authors suggest that the variants lead to haploinsufficiency. However, other mechanisms could be possible. They should discuss the possibility of other outcomes in the discussion section.
5) The authors do not acknowledge the fact experiment in Figure 4 has been partially done previously (Nadiri et al., 2006, Cell and Tissue Research). The authors need to add reference to this paper in the discussion section.
Author Response
Dear reviewer2:
On behalf of our co-authors, we thank you very much for giving us the opportunity to revise our manuscript. We appreciate you for positive comments and constructive suggestions on our manuscript ijms-2039845 entitled “BMPR2 variants underlie nonsyndromic oligodontia”. We have studied all your comments carefully, and the modifications in the revised manuscript were in track mode. We hope this revision could merit publication considering the careful modifications that we made. The point-by-point responses are provided below.
Point 1: In line 94-95, the authors state that their findings suggest that BMPR2 is involved in bone formation. However, this has already been shown previously (e.g., Lowery et al., 2015, J Cell Sci.). This sentence needs to be rewritten to incorporate these previous findings.
Response 1: Thank you for the observation and professional comments. We totally agree with it and have already added the details in “Results” part of our revised paper.
Point 2: Text in Figures 1 and 3 are too small when figures are printed. The font size needs to be increased.
Response 2: Thank you for your careful suggestion. We have performed corresponding changes in our revised Figures 1 and 3.
Point 3: In lines 155-156, the authors use an acronym (BKOTCs) to describe the BMPR2 knockout cell line. This was confusing at first because this acronym was not described previously. I would rather write down everything without using an acronym.
Response 3: Thank you for the constructive comment. We have modified all acronym “BKOTCs” to the full name “BMPR2-knockout 293T cells” in “Results” section of the revised manuscript.
Point 4: In lines 172-173, the authors suggest that the variants lead to haploinsufficiency. However, other mechanisms could be possible. They should discuss the possibility of other outcomes in the discussion section.
Response 4: Thank you for the professional comments. We totally agree with it. According to an analysis of human disease genes, approximal 50% of the mutations in membrane receptors were dominant (Jimenez-Sanchez et al., 2001, Nature). Genetic dominance commonly results from dominant-negative effect or haploinsufficiency. Dominant-negative effect refers to a defective subunit adversely affecting the activity of the normal one. Whereas, haploinsufficiency describes a situation of reduced activity by a heterozygous null allele (Veitia et al., 2017, Clin Genet). It is sometimes difficult to determine a priori whether haploinsufficiency or dominant-negative effect should be responsible for the phenotypes, particularly in the case of intragenic point mutations. In this study, we analysed the expression of BMPR2 by western blotting, and demonstrates that the inactivated BMP/phospho-SMAD1/5/8 signaling were caused by the novel BMPR2 heterozygous variants, suggesting that the effects of BMPR2 are allele dose-dependent. Further results showed that comparing with wild-type group, the activities of phospho-SMAD1/5/8 in heterozygous variants groups were reduced due to heterozygous null allele, but no less than one normal allele group (simulated by 0.5 µg wild-type group). These findings indicate that a haploinsufficiency effect contributes to the genetic mechanism of BMPR2-associated nonsyndromic oligodontia, rather than dominant-negative effect. We have added the above to the “Discussion” section.
Point 5: The authors do not acknowledge the fact experiment in Figure 4 has been partially done previously (Nadiri et al., 2006, Cell and Tissue Research). The authors need to add reference to this paper in the discussion section.
Response 5: Thank you for your constructive advice. We have added the corresponding reference and descriptions in the “Discussion” part.
Round 2
Reviewer 1 Report
As the authors did not perform the suggested experiments of Figure 3, it is difficult to interpret Fig. 3 C-F. Again, when the controls don’t behave, this reviewer cannot agree with the conclusion of Figure 3.
Round 3
Reviewer 1 Report
The authors have addressed the issue raised by this reviewer.